# Mechanosensitive non-equilibrium supramolecular polymerization in closed chemical systems

Xianhua Lang [1,5], Yingjie Huang [1,5], Lirong He[1], Yixi Wang[2], Udayabhaskararao Thumu[2], Zonglin Chu[3], Wilhelm T. S. Huck [4] & Hui Zhao [1] ✉

Chemical fuel-driven supramolecular systems have been developed showing out-of-equilibrium functions such as transient gelation and oscillations. However, these systems suffer from undesired waste accumulation and they function only in open systems. Herein, we report non-equilibrium supramolecular polymerizations in a closed system, which is built by viologens and pyranine in the presence of hydrazine hydrate. On shaking, the viologens are quickly oxidized by air followed by self-assembly of pyranine into micrometer-sized nanotubes. The self-assembled nanotubes disassemble spontaneously over time by the reduced agent, with nitrogen as the only waste product. Our mechanosensitive dissipative system can be extended to fabricate a chiral transient supramolecular helix by introducing chiral-charged small molecules. Moreover, we show that shaking induces transient fluorescence enhancement or quenching depending on substitution of viologens. Ultrasound is introduced as a specific shaking way to generate template-free reproducible patterns. Additionally, the shake-driven transient polymerization of amphiphilic naphthalenetetracarboxylic diimide serves as further evidence of the versatility of our mechanosensitive non-equilibrium system.

Supramolecular polymers[1-5] refer to molecular self-assembly into larger, more complex structures driven by reversible non-covalent interactions. This field has attracted continuous attention as these supramolecular structures could endow materials with unique properties, such as self-healing[6,7] and responsivity[8-10]. Most synthetic supramolecular polymers are formed under equilibrium or as metastable (kinetically trapped) states, whereas living organisms use biofuel-driven dissipative supramolecular structures far from equilibrium to perform complex biological functions. For example, guanosine-5′-triphosphate (GTP) was used as fuel to power transient microtubes for controlling cellular machinery. Inspired by nature's out-of-equilibrium self-assembled supramolecular systems, researchers developed chemical reaction cycles to drive assembly and feedback-controlled disassembly of small molecules or nanoparticles[11,12]. Various high-energy chemicals have been used as fuels in chemical reaction cycles for dissipative self-assembly, such as EDC[13-15], DNA[16,17], amino acid[18,19], ATP[20-22], light[23-27], and other chemicals[28-32]. The development of fuel-driven dissipative self-assembly opened broad perspectives for smart materials with life-like features, such as oscillations[33-37], replications[38-40], transient nanostructures[18], dissipation[41-43], and adaption[44]. However, these chemical systems typically consume all their 'fuel' within a single cycle, and repeated operation fuel additions lead to the accumulation of waste. To date, only the well-known Belousov-Zhabotinsky and a few related oscillators can show oscillations in a closed system—but there are no design rules that allow a rational design of fuel-driven closed systems.

[1]School of Chemical Engineering, State Key Lab of Polymer Materials Engineering, Sichuan University, 610065 Chengdu, China. [2]Institute of Fundamental and Frontier Sciences, University of Electronic Science and Technology of China, 610054 Chengdu, China. [3]College of Chemistry and Chemical Engineering, Hunan University, 410082 Changsha, China. [4]Institute for Molecules and Materials, Radboud University, Nijmegen, The Netherlands. [5]These authors contributed equally: Xianhua Lang, Yingjie Huang. ✉e-mail: zhaohuichem@scu.edu.cn

Here, we overcome above mentioned difficulties by creating a dissipative supramolecular polymerization from a gas-liquid two-phase redox reaction system. In our proposed closed system, the chemical fuel, oxygen, is stored in the gas phase while inactive monomers and hydrazine are stored in the aqueous solution. Diffusion of aerial oxygen into the liquid phase is accelerated by shaking, which instantly leads to oxidation of viologens which subsequently co-assemble with pyranine into supramolecular nanotubes. Without mechanical agitation, the diffusion of aerial oxygen into the liquid phase slows down and reduction of viologens by hydrazine present in the liquid phase becomes the dominating reaction, leading to disassembly of supramolecular polymers into inactivated monomers. Of particular importance is the waste, nitrogen gas, which is released from solution in each cycle, ensuring that no waste accumulates after each reaction cycle.

Moreover, our proposed non-equilibrium system in which mechanical force modulated chemical fuel incorporation is more attractive because of its convenience and shorter action time as against the usual chemical fuel-driven non-equilibrium systems. So far, mechanical force-driven responsive supramolecular self-assembly[45] as well as transient self-assembly[46] have been reported. More interestingly, regular sound wave vibrations as a specific mechanical force have been applied to construct non-equilibrium redox reactions[47–49], but no work has been reported on the non-equilibrium supramolecules

self-assembly driven by high-frequency vibrations such as ultrasound. In this work, we have employed a novel approach involving ultrasound instead of vibration to achieve reconfigurable and regular heart-shape patterns, which were different with regular round cycle pattern generated by low-frequency sound waves. The principle presented here may provide a general and promising route to ultrasound triggered reaction-diffusion patterns based on dissipative supramolecular self-assembly, advancing the development of transient functional materials with life-like behaviors.

## Results

### Shake-induced out-of-equilibrium redox reaction of viologens in open and closed chemical systems

We first investigated the dissipative performance of viologen[28,47], which is the central building block for our proposed dissipative supramolecular polymerization, in both open and closed chemical systems (Fig. 1a). The dissipative reaction cycles are driven by shaking-induced redox reactions in an air-aqueous two phase, in which $O_2$ is the oxidant and $N_2H_4 \cdot H_2O$ is the reductant. The first experiment was done with 3 mM of alkyl substituted viologen ($C_{12}$-MV$^{\cdot+}$) at pH = 12 in the presence of $N_2H_4 \cdot H_2O$ (10% v/v). A decayed color change ($k_{reduction}$ = $6.18 \times 10^{-4}$ L $\cdot$ mol$^{-1} \cdot$ s$^{-1}$, $k_{oxidation}$ = $1.50 \times 10^4$ L $\cdot$ mol$^{-1} \cdot$ s$^{-1}$) was observed during the shake-induced redox reaction of $C_{12}$-MV$^{2+}$, as shown in Fig. 1b and Supplementary Movie 1. Time-dependent UV-vis spectra

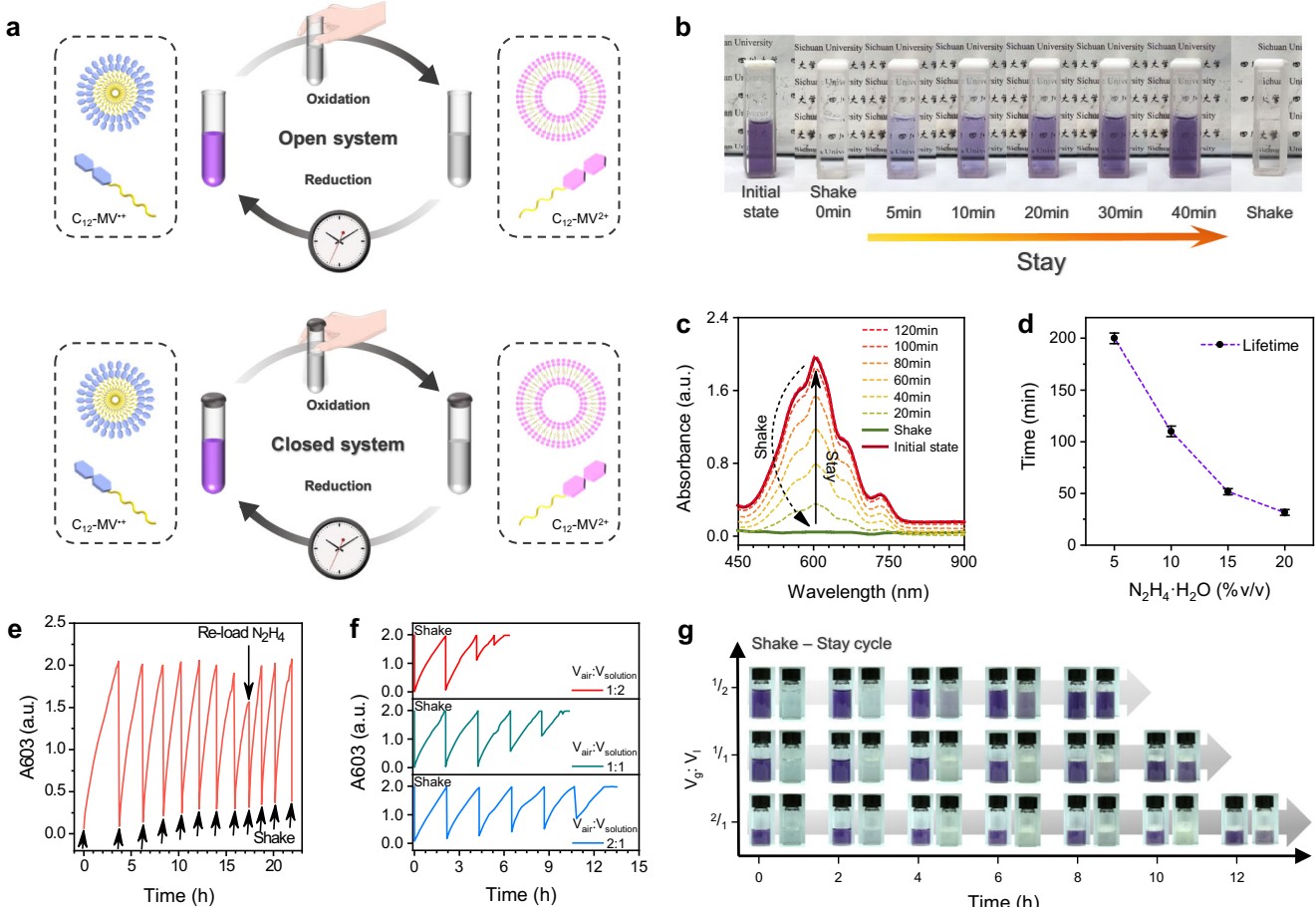

**Fig. 1 | Shake-induced out-of-equilibrium redox reaction of C$_{12}$-MV$^{\cdot+}$ in an open and closed system. a** Schematic illustration of shake-triggered redox of C$_{12}$-MV$^{\cdot+}$ in an open and closed system. **b** Photos for the solution of the shake-induced transient redox reaction. **c** Time-dependent UV-vis spectra depicting redox behavior of C$_{12}$-MV$^{\cdot+}$ (3 mM) triggered by shaking in a buffer solution containing reducing agent (10% v/v). **d** Lifetime of shake-induced transient redox reaction in different amounts of N$_2$H$_4 \cdot$ H$_2$O. Data are presented as the average values ± s.d. (n = 3).

**e** Repeated cycles of reduction-oxidation process of C$_{12}$-MV$^{\cdot+}$ (3 mM) powered by hand-shake in the presence of N$_2$H$_4 \cdot$H$_2$O (5% v/v), reloading (5% v/v). $k_{reduction}$ = $6.18 \times 10^{-4}$ L mol$^{-1}$ s$^{-1}$, $k_{oxidation}$ = $1.50 \times 10^4$ L mol$^{-1}$ s$^{-1}$. **f** Temporal evolution of absorbance intensity at 603 nm for shake-induced redox reaction in a closed system. **g** Visualization of chemochromism under different volume ratios of air-to-aqueous solutions for the repeated shake-stay cycles in a closed system, [C$_{12}$-MV$^{\cdot+}$] = 3 mM, [N$_2$H$_4 \cdot$ H$_2$O] = 10% (v/v).

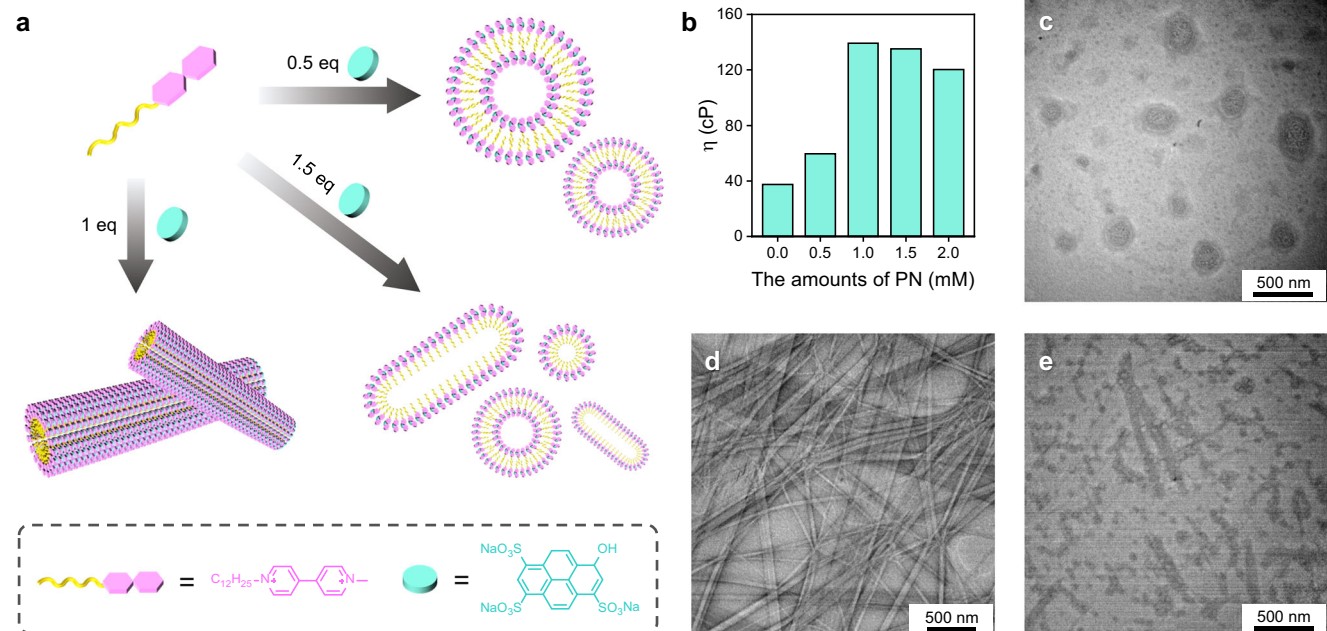

**Fig. 2 | Characteristics of structure-controlled supramolecular polymerization. a** Schematic diagram of structure-controlled self-assembly dependent on the ratios of $C_{12}$-$MV^{2+}$ and PN. **b** Viscosity of different ratios of $C_{12}$-$MV^{2+}$ and PN solutions. TEM images depicting the self-assembly morphology resulting from varying amounts of PN added to $C_{12}$-$MV^{2+}$ solution, 0.5 mM PN (**c**), 1 mM (**d**), and 1.5 mM (**e**). [$C_{12}$-$MV^{2+}$] = 1 mM.

were recorded as shown in Fig. 1c. Upon shaking a sudden disappearance of absorbance intensity at 603 nm (corresponding to $C_{12}$-$MV^{\cdot+}$) was seen, and then the absorption band gradually increased, signifying the decay phase. The lifetime of the shake-induced transient redox reaction of $C_{12}$-$MV^{\cdot+}$ is highly dependent on the number of reductants. As the equivalent of $N_2H_4 \cdot H_2O$ increased from 5% to 20% (v/v), it was observed that the lifetime of the transient $C_{12}$-$MV^{2+}$ drastically decreased from 300 min to 35 min (Fig. 1d and Supplementary Fig. 15), which is because the high concentration of the reducing agent resulted in a faster reduction of $C_{12}$-$MV^{2+}$. The conversion associated with the color change mechanism was further investigated by time-dependent proton NMR. The addition of $N_2H_4 \cdot H_2O$ into the solution of $C_{12}$-$MV^{2+}$ (pH = 12) induced the gradual disappearance of peaks at 8.5 and 9.1 ppm corresponding to the protons in $C_{12}$-$MV^{2+}$ over time, indicating the formation of radicals ($C_{12}$-$MV^{\cdot+}$) (Supplementary Fig. 16). The corresponding signals (8.5 and 9.1 ppm) reappeared upon shaking the sample in the NMR tube. Additionally, EPR experiments on the basic solution of $C_{12}$-$MV^{\cdot+}$ in the presence of the excess amount of $N_2H_4 \cdot H_2O$ also supported the transient formation of $C_{12}$-$MV^{2+}$: signal (345 mT) corresponding to $C_{12}$-$MV^{\cdot+}$ disappeared by shaking and gradually increased back (Supplementary Fig. 17). To demonstrate the repeatability of this transient system, refueling by subsequent shake was carried out and multiple cycles were obtained, in which the number of cycles could be further increased after reloading of reductant (Fig. 1e).

Subsequently, we capitalized on the low solubility of air in water to devise a dissipative reaction that can be initiated by shaking within a closed system. UV-vis measurements were performed to investigate the reaction process. As displayed in Fig. 1f, the UV-vis spectra exhibited shake-induced reversible cycles (disappear and gradually raise of absorption band at 603 nm) which is similar to the open system described above. However, compared to the open system, the volume ratio of gas-to-liquid ($V_g/V_l$) has a tremendous effect on the repeatability of these cycles since the air is fixed and limited in a closed system. When $V_g/V_l = 1:2$, the redox reaction cycles could only be repeated two times and additional shake did not trigger the redox reaction of $C_{12}$-$MV^{2+}$, which attributed to the complete consumption of

oxygen (fuel) in the closed system. As the volume ratio of gas-to-liquid increased from 1:2 to 2:1, it was observed that the repeated dissipative process of $C_{12}$-$MV^{2+}$ increased from 2 times to 6 times. Further evidence for these results was provided by the visualization of multiple cycles of transient discoloration driven by shaking in a closed system (Fig. 1g). It was worth noting that the redox cycles could continue by simply introducing fresh air, in which $O_2$ was reloaded as the fuel in the gas phase (Supplementary Fig. 18).

## Shake-induced transient supramolecular polymerization

We constructed the mechanosensitive supramolecular self-assembly system based on the study of thermodynamically stable supramolecular polymerization of alkane substituted viologen ($C_{12}$-$MV^{2+}$) and pyranine (PN), in which the self-assembly was driven by charge transfer interaction (CT interaction) and amphiphilic interaction[50]. Moving ahead we wanted to analyze the structure controlled by the ratio of incoming PN to $C_{12}$-$MV^{2+}$ (Fig. 2a). By adding PN to the $C_{12}$-$MV^{2+}$ solution (1 mM), we observed a visible growth of the self-assembled structure according to the viscosity tests as shown in Fig. 2b, and the extent of growth depended on the concentration of incoming PN. However, adding too much PN leads to a decrease in solution viscosity which is caused by the disruption of the self-assembled structure. The effect on the controllable structure could be further investigated by TEM tests. When 0.5 equivalents of PN were added to the $C_{12}$-$MV^{2+}$ solution, vesicles with an average diameter of 300 nm were formed (Fig. 2c), while the addition of 1 equivalent of PN resulted in the formation of one-dimensional structures with a diameter of 80 nm (Fig. 2d). However, the addition of 1.5 equivalents of PN resulted in the disruption of the fibers (Fig. 2e). These observations are crucial for the development of chemically fuel-driven assemblies.

Moreover, a shake-driven transient supramolecular self-assembly system was constructed and the mechanism of transient polymerization was shown in Fig. 3a, which was similar to the monomer $C_{12}$-$MV^{\cdot+}$. During the process of transient supramolecular polymerization, shake is the key trigger for accelerating the diffusion of $O_2$ from air into the solution. Upon shaking, similar to the individual monomer, the $C_{12}$-$MV^{\cdot+}$ turned to $C_{12}$-$MV^{2+}$ by oxidation and followed by co-assembly

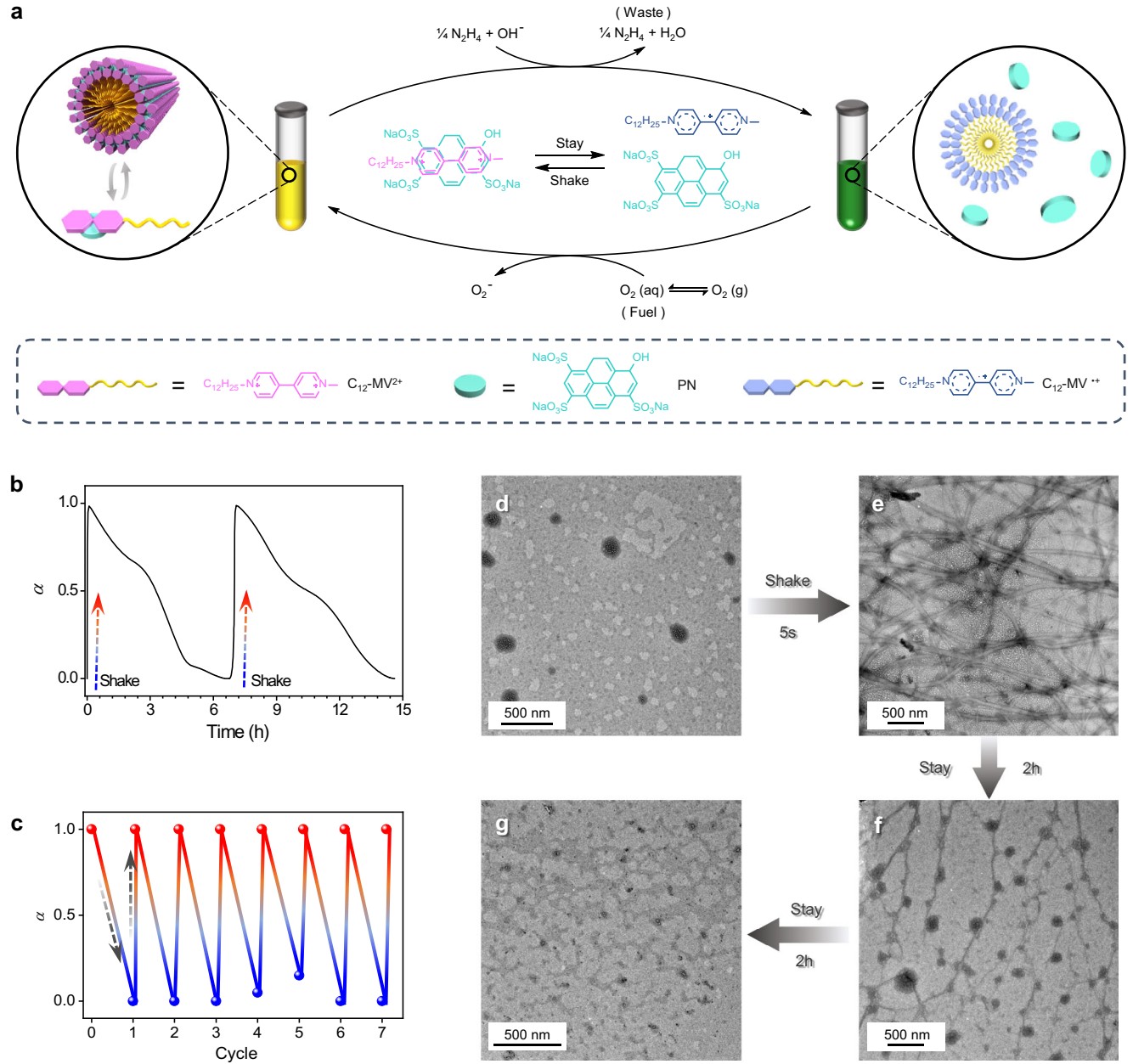

**Fig. 3 | Characteristics of shake-induced transient supramolecular polymerization. a** Schematic representation of dissipative supramolecular polymerization based on $C_{12}$-$MV^{2+}$ and PN triggered by shake. **b** Time-dependent evolution of viscosity trend depicting the kinetics of transient supramolecular polymerization, $C_{12}$-$MV^{2+}$ (3 mM) and PN (3 mM) in the presence of $N_2H_4 \cdot H_2O$ (10% v/v). **c** Polymerization-depolymerization cycles of $C_{12}$-$MV^{2+}$ and PN driven by shake. **d**–**g** TEM images of solution with $C_{12}$-$MV^{2+}$/ PN (1 mM/ 1 mM) in the presence of $N_2H_4 \cdot H_2O$ (10% v/v) before and after shake over time. $\alpha$ is the extent of self-assembly of $C_{12}$-$MV^{2+}$/PN.

with pyranine into supramolecular polymers. After shake is stopped, $C_{12}$-$MV^{2+}$ gradually turned back $C_{12}$-$MV^{\cdot+}$ by reduction in the presence of excess amount of $N_2H_4 \cdot H_2O$, which led to the decayed disassembly of $C_{12}$-$MV^{2+}$/ PN supramolecular polymers. Here too, nitrogen gas evolved as the waste but would not accumulate in the reaction solution because of its low solubility in water, which endowed a unique property, waste traceless, with our proposed system (Supplementary Fig. 19). The kinetics of transient polymerization were characterized by viscosity measurement and TEM. As shown in Fig. 3b, the degree of self-assembly ($\alpha$) [51], which could be confirmed according to Eq. (1), after shaking in the solution containing $C_{12}$-$MV^{2+}$ (3 mM) and PN (3 mM) in the presence of $N_2H_4 \cdot H_2O$ (10% v/v) decreased (disassemble) gradually over 7 h, but another shake triggered another cycle of supramolecular polymerization. The reversible nature of

supramolecular polymerization-depolymerization was displayed by performing at least six cycles in a closed system (Fig. 3c). The number of cycles of shake-induced supramolecular polymerization could be further increased by the re-addition of $N_2H_4 \cdot H_2O$ and fresh air. To monitor the morphological changes of transient polymerization in real-time, the system was characterized using TEM at different time points (Fig. 3d–g). TEM images showed shake-triggered molar equivalent $C_{12}$-$MV^{2+}$ and PN (1 mM) self-assembled into well-defined nanofibers with 100 nm in diameter and several hundreds of microns in length. Subsequently, the fibers gradually shrunk and transformed into short fibers and vesicles when the solution was kept stagnant. The fibers reappeared after mechanical stimulation (Supplementary Figs. 20 and 21). The evolution in structure of supramolecular polymers over time seen from TEM is in line with the above viscosity

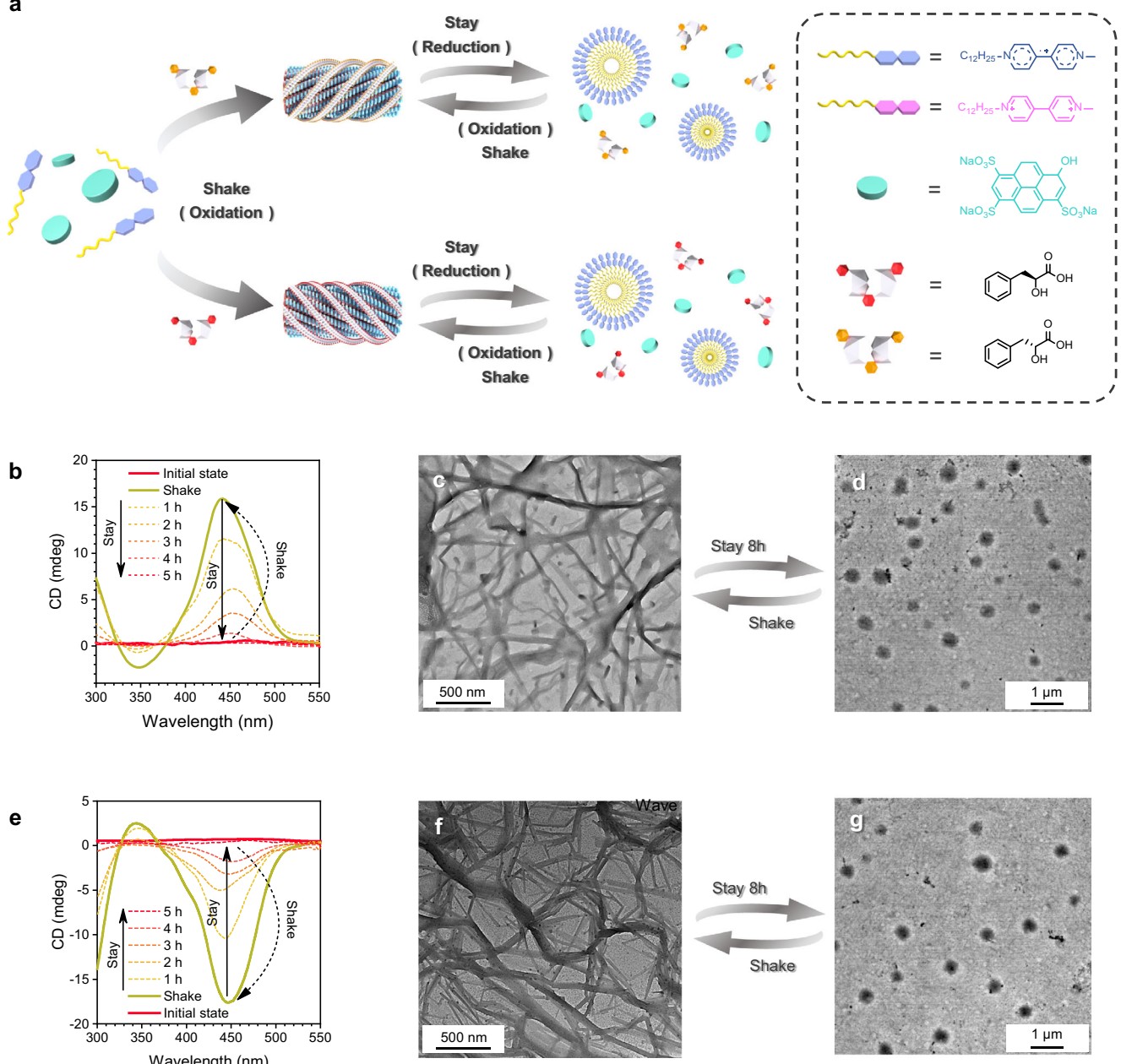

**Fig. 4 | Shake-induced transient supramolecular chirality. a** Schematic representation of transient supramolecular chirality polymer triggered by hand-shake. Time-dependent changes in circular dichroism signal intensity demonstrating **b** *L*-(-)-Phenyllactic acid or **e** *D*-(+)-Phenyllactic acid led to a temporal helical polymer. TEM images show the structure of helix supramolecular assembly formed in the presence of **c** *L*-(-)-Phenyllactic acid or **f** *D*-(+)-Phenyllactic acid. TEM images show the morphology after dissipation of $C_{12}$-MV$^{•+}$/ PN in the presence of **d** *L*-(-)-Phenyllactic acid or **g** *D*-(+)-Phenyllactic acid.

measurements. These results provide clear evidence of dynamic supramolecular polymerization.

### Transient supramolecular helix

The above design of charged supramolecular polymer also offers additional electrostatic interactions, which could accommodate host-guest interactions for the formation of transient chiral supramolecular structures via introduction of charged chiral molecule into the reaction solution. To verify this hypothesis, chiral *L*-(-)-Phenyllactic acid and *D*-(+)-Phenyllactic acid were introduced as an additive in two separate dissipative processes, respectively (Fig. 4a). Circular dichroism (CD) was used to measure the time-dependent variation in the chirality of supramolecular polymers. Interestingly, a strong CD signal ($λ_{max} = 450$ nm) was observed when the aqueous solution with

$C_{12}$-MV$^{2+}$, PN, *L*-(-)-Phenyllactic acid and $N_2H_4 \cdot H_2O$ upon shaking (Fig. 4b), indicating the formation of single handed supramolecular helix by the addition of *L*-(-)-Phenyllactic acid into supramolecular polymerization system. The CD signals gradually disappeared as disassembly of $C_{12}$-MV$^{2+}$/ PN based supramolecular polymer by reduction of $N_2H_4 \cdot H_2O$. Similarly, a shake-induced transition supramolecular helix of opposite handed helix is formed when *D*-(+)-Phenyllactic acid is introduced into supramolecular polymerization system (Fig. 4e). Additionally, supramolecular helix was also characterized by TEM. Twisted micrometers long fibers instead of nanotubes were formed when solutions ($C_{12}$-MV$^{2+}$/ PN, $N_2H_4 \cdot H_2O$) with additional chiral *L*-(-)-Phenyllactic or *D*-(+)-Phenyllactic acid were shaken (Fig. 4c, f). In the presence of hydrazine hydrate, the supramolecular helical structure gradually decays into helices of 400 nm in length until vesicle

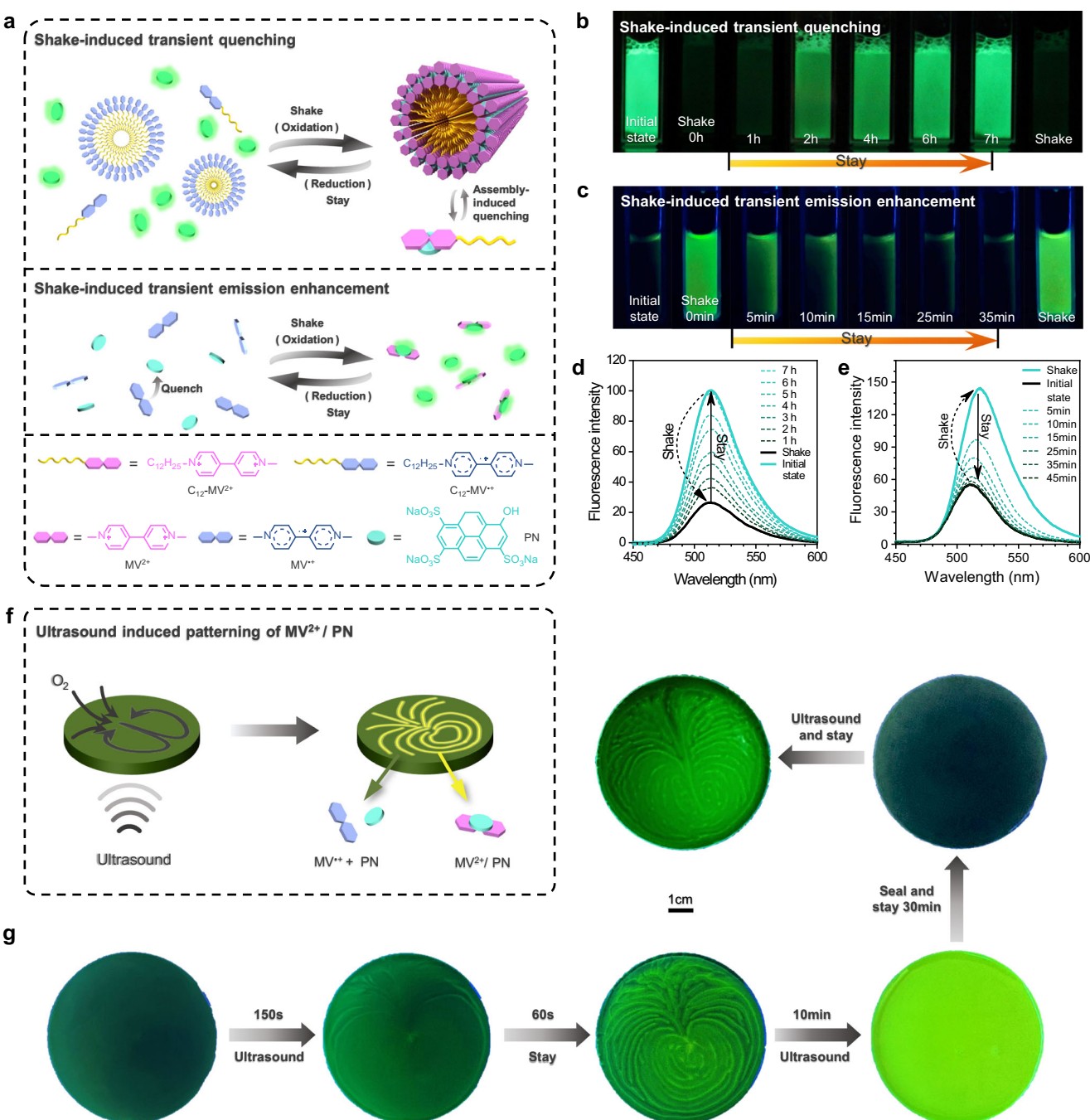

**Fig. 5 | Transient fluorescence and ultrasound guided patterns. a** Schematic illustration of shake-triggered transient fluorescence. **b, c** Photos of shake-induced quenching and emission enhancement. **d** Fluorescence spectra of shake-induced quenching. 0.2 mM $C_{12}$-$MV^{2+}$, 0.1 mM PN, and $N_2H_4 \cdot H_2O$ 5% (v/v). **e** Fluorescence spectra of shake-induced emission enhancement. 2 mM $MV^{2+}$, 0.2 mM PN, and $N_2H_4 \cdot H_2O$ 20% (v/v). **f** Schematic illustration of ultrasound-induced oxygen directed diffusion to form patterns. **g** Ultrasound-induced patterns for $MV^{2+}$/PN, under UV light (365 nm).

formation when the solution stayed still over time (Fig. 4d, g and Supplementary Figs. 22 and 23), which was consistent with CD measurement.

### Transient fluorescence and ultrasound guided patterns

EPR measurement on equimolar $C_{12}$-$MV^{2+}$/PN (1 mM) samples in the presence of $N_2H_4 \cdot H_2O$ (20% v/v) was employed to characterize the dynamic behavior of radical formation ($C_{12}$-$MV^{\cdot+}$) during the disassembly process which eventually influences the supramolecular polymerization. The EPR spectrum of the solution measured immediately after the shaking show the quenching of radical signal at 345 mT

and the signal is gradually increased over time while the solution stands still during the measurement (Supplementary Fig. 24).

In general, the radicals of viologens known to quench on fluorescent dyes, rather, in our case the solution of $C_{12}$-$MV^{\cdot+}$/PN showed stronger emission than $C_{12}$-$MV^{2+}$/PN (Supplementary Table 1, fluorescent quantum yield $\varphi_f$ : 1.43% vs 0.29%). This is because $C_{12}$-$MV^{\cdot+}$ itself tended to form nanoaggregates and is likely to precipitate from the solution, which leaves little opportunity for energy transfer between $C_{12}$-$MV^{\cdot+}$ and PN (Fig. 5a). Moreover, the fluorescence intensity of the solution was rapidly quenched upon shaking the solution for few seconds, which was the result of aggregation caused quenching

(ACQ) of PN during the supramolecular polymerization process[48] (Fig. 5b, d, Supplementary Figs. 25, and 26, and Supplementary Movie 2). Fluorescence of the solution gradually turned on when stayed it still over time, which indicated the system had unique shake-induced quenching properties. Strikingly, in contrast to the $C_{12}$-$MV^{2+}$/ PN solution, the $MV^{2+}$/ PN exhibited behavior of shaking-induced fluorescence enhancement (Fig. 5a and Supplementary Movie 3). In the system of $MV^{\cdot+}$/ PN/ $N_2H_4 \cdot H_2O$, the remarkable solubility and non-aggregating property of $MV^{\cdot+}$ facilitate a robust interaction with PN, leading to a significant energy transfer and consequent amplification of fluorescence quenching. And the fluorescence intensity of solution of $MV^{\cdot+}$/ PN/ $N_2H_4 \cdot H_2O$ raised rapidly after shaking the sample for 2 s (Fig. 5c, e and Supplementary Fig. 9), which is because the reduced energy transfer between $MV^{\cdot+}$ and PN when the radical $MV^{\cdot+}$ were turned to $MV^{2+}$ by oxidation of $O_2$ (Supplementary Fig. 30).

Next, we sought to produce a naked-eye visible transient fluorescent pattern inspired by the above shake-induced fluorescence cycles of quenching and enhancement. First, a sonication linked template was utilized to generate patterns on the solution of ($MV^{2+}$ or $MV^{\cdot+}$)/PN/$N_2H_4 \cdot H_2O$, unfortunately, which could not produce patterns because the patterns were destroyed by template when it was taken away from the aqueous solution. Alternatively, inspired by Kim et al. use low-frequency-acoustic-vibration to produce transient patterns[49], we used the ultrasound (an untouched shake) to drive the mechano-sensitive redox reaction for pattern formation (Supplementary Figs. 34–50 and Supplementary Movie 4–7). Our research has unveiled ultrasonic vibration as an unconventional method for patterning that employs a unique imaging principle, diverging from that of low-frequency vibrations (Supplementary Figs. 42 and 43).

Ultrasound is mechanical waves that propagate in an elastic medium, dissipative solution in our case, which accelerates the diffusion of oxygen from air to aqueous solution at specific regions. We observed that ultrasound wave in the reaction medium affects the diffusion rate of the gas in a certain direction of the solution, and subsequently induces a directional oxidation process which results in unique visible patterns (Fig. 5f and Supplementary Figs. 43b and 44). Depending on the viscosity of the medium, different patterns under visible light can be obtained. Take the system ($MV^{\cdot+}$/PN/$N_2H_4 \cdot H_2O$) as a model, the solution exhibited islands when the solution was subjected to sonication for 2.5 min and stayed for 1 min. The patterns of islands gradually changed to concentric circles of diffusion patterns when the solution was loaded with more PEG (10 kDa, 0–5%, weight contents) (Supplementary Figs. 34 and 35 and Supplementary Movie 4). The changes in the pattern are due to the variation in solution viscosity with increased as loading more PEG, which reduces the diffusion rate of oxygen and makes its oxidation flow more continuous. A continuous flow pattern can be obtained under sonication using $C_{12}$-$MV^{2+}$ alone without the addition of PEG due to the surface activity of $C_{12}$-$MV^{2+}$. Moreover, the size of the central pattern is directly related to the concentration of $C_{12}$-$MV^{2+}$ which affects the surface tension of the solution (Supplementary Figs. 36 and 37 and Supplementary Movie 6). However, during the attempted patterning of $C_{12}$-$MV^{2+}$/ PN, the aggregated $C_{12}$-$MV^{\cdot+}$ was difficult to be dispersed again by ultrasound and stable patterns could not be obtained (Supplementary Figs. 39 and 40). Best patterns also can be seen under both visible and UV (365 nm) light since the solution ($MV^{\cdot+}$/PN/$N_2H_4 \cdot H_2O$) had shaken-induced emission enhancement properties. Especially, a clearly reproduced concentric heart shape pattern was generated when the solution was on sonication for around 2 min and stayed 1 min. When the solution was on sonication, ultrasound induced liquid flow was accompanied by directional diffusion of oxygen to oxidize $MV^{\cdot+}$/PN for discoloration, which led to the formation of patterns (Fig. 5f and Supplementary Movie 5). Furthermore, we observed the formation of different degrees of heart-shaped patterns at distinct stages of diffusion during the continuous ultrasound process

(Supplementary Fig. 46 and Supplementary Movie 7). The patterns spread throughout the surface of the solution when the sonication for longer time (10 min) leading to bright green fluorescent from the entire solution.

To study our ultrasound triggered patterns more in depth, we compared parallelly our system with Kim's work on audiosound triggered patterns. Audio-speaker and ultrasonic equipment both generate sound wave, however, there are several different points when low-frequency acoustic waves and ultrasound are used separately as a trigger for the generation of reproducible spatiotemporal patterns of viologens. (1) Patterns are different: regular round circles in Kim's work (based on low-frequency sound wave vibrations)[47] while regular heart shape in our work (based on ultrasonic equipment). (2) Proposed mechanism is different: patterning formation is due to different concentrations of oxygen at the antinodal and nodal positions on the water surface based on audio-speaker[47,48], therefore, the formation of pattern is a replication of the vibrational ripples on the surface of the solution (Supplementary Figs. 42 and 43a), while the patterning formation under ultrasonic mode is due to the induced directional diffusion of oxygen, the diffusion direction is shown in Supplementary Figs. 43b and 44. (3) Viscosity dependence is different: heart-shaped patterns under ultrasound can be resized, unlike regular circles in low-frequency sound waves which are not adjustably resizable by altering the viscosity (Supplementary Fig. 45). We have observed that increasing viscosity can enhance the continuity of ultrasonically induced patterns and enable better control over the size of heart-shaped patterns. According to Newton's law of viscosity, the magnitude of viscosity is proportional to the magnitude of shear stress (internal friction). With the power of ultrasound being constant and the driving force unchanged, an increase in resistance reduces the range of influence of central diffusion flow on the surrounding solution.

The reversibility of this dark-fluorescent patterns-fully bright cycle depends on whether the system is opened or sealed. The fully bright green fluorescent-on state would turn back to dark state once the solution was sealed and stayed still for around 30 min (Fig. 5g). However, in the case of the system exposed to the air, it turned to bright island-type patterns (Supplementary Fig. 41) when the solution was kept still for about 30 min, and then gradually turned to bright green throughout of surface instead of dark state when keeping the solution for a longer time. Interestingly, the shape of mold affects the directional pattern. The mold changed from round to triangle or square shape mold for ultrasound also induced transient patterning but not equivalent to the circular shape. It turned out that only islands or short lines patterns formed (Supplementary Figs. 49 and 52) when the solution containing $C_{12}$-$MV^{\cdot+}$/PN/$N_2H_4 \cdot H_2O$ in triangle or square shape mold, which was the result of mechanic waves from ultrasound took place different reflections on different shapes of molds.

## Shake-induced transient supramolecular polymerization of $C_{12}$-NDI-PEG350

To verify the generality of our design concept, we constructed another shake-driven supramolecular self-assembly system based on the reversible redox properties of naphthalenetetracarboxylic diimide (NDI). The amphiphilic compound $C_{12}$-NDI-PEG350 was prepared by a statistical condensation reaction of 1,4,5,8-naphthalenetetracarboxylic dianhydride with hydrophilic polyethylene glycol groups amine and hydrophobic dodecyl amine. As illustrated in Fig. 6a, the mechanism of transient polymerization of $C_{12}$-$NDI^{2-}$-PEG350 was similar to the $C_{12}$-$MV^{\cdot+}$/PN. Upon shaking, $C_{12}$-$NDI^{2-}$-PEG350 is oxidized by oxygen in the air to form $C_{12}$-NDI-PEG350, leading to the formation of supramolecular polymers through self-assembly. In the presence of excess hydrazine hydrate, $C_{12}$-NDI-PEG350 gradually returned to its reduced form ($C_{12}$-$NDI^{2-}$-PEG350), accompanied by the release of nitrogen as a waste product. The color change as time shown in Fig. 6b effectively

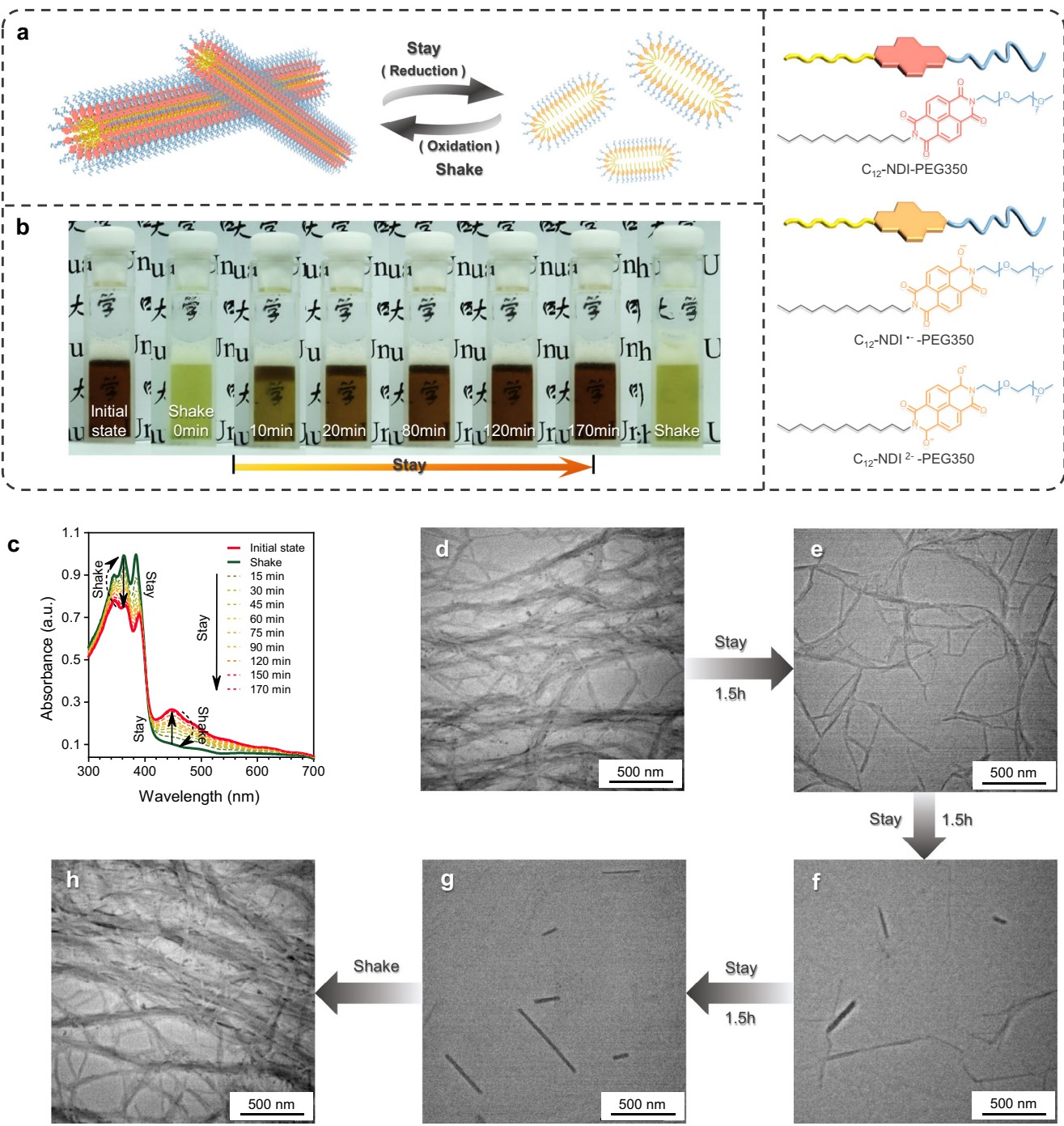

**Fig. 6 | Shake-induced out-of-equilibrium redox reaction of C$_{12}$-NDI-PEG350 in closed system. a** Schematic representation of dissipative supramolecular polymerization based on C$_{12}$-NDI-PEG350 triggered by shake. **b** Visualization of chemochromism of the solution indicating the shake-induced transient redox reaction, [C$_{12}$-NDI-PEG350] = 1 mM, [N$_2$H$_4$ · H$_2$O] = 10% (v/v). **c** Time-dependent UV-vis spectra depicting redox behavior of C$_{12}$-NDI-PEG350 (1 mM) triggered by shaking in a buffer solution (pH = 8) containing reducing agent (10% v/v). **d–h** TEM images of solution C$_{12}$-NDI-PEG350 (1 mM) with the presence of N$_2$H$_4$ · H$_2$O (10% v/v) before and after shake over time.

demonstrated the shake-driven reversible redox reaction of C$_{12}$-NDI$^{2-}$-PEG350. We then investigated the kinetics of transient supramolecular polymerization by UV-vis spectra and TEM. The time-dependent UV-vis spectra showed that shaking resulted in a sudden decrease in the absorption intensity at 425 nm (corresponding to C$_{12}$-NDI$^{2-}$-PEG350) and a corresponding increase at 380 nm (corresponding to C$_{12}$-NDI-PEG350), followed by a gradual increase in the absorption peak at 425 nm and a gradual decrease in the absorption peak at 380 nm, marking the decay phase (Fig. 6c). Moreover, TEM was utilized to

observe the morphological changes that occurred during transient polymerization, as depicted in Fig. 6d–h. TEM images indicated that shake-driven C$_{12}$-NDI-PEG350 self-assembled into nanofibers with a diameter of 50 nm. However, the nanofibers gradually collapsed and transformed into short fibers and finally into nanorods when the solution stayed stationary. Notably, the supramolecular polymer was able to regenerate upon being subjected to new mechanical stimulation. These results validate the broader scalability of our engineered mechanosensitive system.

## Discussion

In conclusion, we have demonstrated a dissipative supramolecular polymerization in closed chemical systems driven by mechanical force-shake, which endows transient supramolecular polymer with a kinetically controlled ability. The lifetime, structure, and function of supramolecular polymer are temporally controlled. The dissipative system could be repeated several times in closed chemical systems without the need for additional loading of fuels. Particularly, nitrogen gas as the side product of the reaction cycles immediately leaves the solution resulting in a traceless waste in the system. The presence of chiral molecule additives in our dissipative system leads to the formation of helix polymer, and the temporal control of assembly demonstrated a transient helicity. Shake-induced fluorescence quenching and emission enhancement were obtained depending on the structures of viologens. Moreover, predictable, transient patterns with fluorescence were generated by using ultrasound as an untouched way of shake to drive the redox reaction of viologens. In addition, the generality of our mechanosensitive non-equilibrium system was displayed by building another shake-driven transient self-assembly system. We believe that this shake-driven out-of-equilibrium supramolecular self-assembly could endow functional materials with unique properties, such as mechanosensitive, transient chirality, transient fluorescence, and template-free patterns, providing promising opportunities for creating life-like novel materials.

## Methods

### Materials

All chemicals were purchased from Bide Chemical Technology Co., Ltd. and used directly without further purification. All the water used to prepare the aqueous solutions was obtained from an ultrapure purification system and had a resistance of 18.2 ($M\Omega$ cm) after deionization.

### Syntheses and characterization

For information on the synthesis procedures and compound details mentioned in this report, please refer to the Supplementary Information Document. Detailed characterization of compounds and measurement protocols have been also provided in Supplementary Information.

### Sample preparation

The tests of the shake-driven non-equilibrium system of $C_{12}$-$MV^{2+}$ were carried out in a buffer solution with pH 12, and the tests for $C_{12}$-DNI-PEG350 were performed in a pH 8 buffer solution. As an example, a shake-driven transient supramolecular polymerization consisting of $C_{12}$-$MV^{2+}$ and PN was used. Stock solutions of $C_{12}$-$MV^{2+}$ (5 mM) and PN (10 mM) were separately prepared in the buffer solution with a pH of 12. To prepare the sample, the required amount of stock solution of $C_{12}$-$MV^{2+}$ was added to a bottle and diluted to the desired concentration with phosphate buffer. Afterward, the required quantities of PN stock solution and hydrazine hydrate were added successively to the above solution under vortex conditions, followed by sonication for 1 min. The solution was then left to stand for a while until complete reduction of $C_{12}$-$MV^{2+}$ to $C_{12}$-$MV^{·+}$ occurred before subsequent testing.

### Definition of the extent of self-assembly

The extent of self-assembly of $C_{12}$-$MV^{2+}$/PN is defined as $\alpha$, which could be confirmed according to Eq. (1). The values of '1' and '0' represent the highest and lowest degree of aggregation, respectively.

$$\alpha = \frac{\eta - \eta_0}{\eta_{max} - \eta_0} \qquad (1)$$

$\alpha$ – the degree of aggregation; $\eta_{max}$ – viscosity at maximum aggregation;

$\eta$ – viscosity at different aggregation degrees; $\eta_0$ – initial viscosity.

### Ultrasound-induced pattern generation experiments

The experiment utilized the $MV^{2+}$-PN as a model to showcase ultrasound-induced transient patterning. Initially, $MV^{2+}$, PN, and PEG were dissolved in a pH 12 buffer solution within a 50 ml centrifuge tube. Subsequently, the solution was supplemented with hydrazine hydrate and thoroughly mixed. After $MV^{2+}$ was fully reduced to $MV^{·+}$, the solution was poured into a mold in the sonication apparatus and a distinct fluorescent heart-shaped pattern appeared after 150 s of sonication and ~60 s of resting, under 365 nm ultraviolet light. Afterward, sonication was continued to allow complete oxidation of the sample. Finally, leaving the solution sealed for 30 min would return it to initial state, while leaving it in the open state would result in a random spot pattern.

## Data availability

All data are available in the main text or the supplementary information. Data is also available from the corresponding author upon request. Source data are provided with this paper.

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

## Acknowledgements

This work is supported by Technology of Sichuan Province under grant No. 2022YFN0060 (H.Z.), Natural Science Foundation of China under grant No. 52103121 (L.H.), NSFC Grant No. 22050410280 (U.T.), Fundamental Research Funds for the Central Universities under YJ202117 (L.H.), and State Key Laboratory of Polymer Materials Engineering (SCU) under grant No. sklpme2020-3-03 (L.H.) and No. sklpme2020-3–14 (H.Z.), Sichuan University-Luzhou Science and Technology Innovation Platform Construction Project Grant No. 2022CDLZ-20 (H.Z.). We thank Yanping Huang from Center of Engineering Experimental Teaching, School of Chemical Engineering, Sichuan University for the help of NMR and TEM measurement.

## Author contributions

H.Z. conceived and supervised the project. X.L., Y.H., and Y.W. carried out experiments. X.L., Y.H., L.H., Z.L.C., and H.Z. carried out the data analysis. X.L., Y.H., U.T., W.T.S.H., and H.Z. wrote the manuscript. All the authors discussed the results and commented on the manuscript.

## Competing interests

The authors declare no competing interests.
