## [Peer Review File · Nature Communications]

Mechanosensitive non-equilibrium supramolecular polymerization in closed chemical systemsReviewers' Comments:

Reviewer #1:

Remarks to the Author:

The authors report the transient self-assembly of alkyl substituted viologen and pyranine using oxygen and hydrazine hydrate as oxidizing and reducing agent, respectively. By shaking the system, they accelerated the diffusion of oxygen from air into solution, resulting in oxidation of viologen and co-assembly. As shaking is stopped, viologen gradually reduced by hydrazine, leading to disassembly. Introducing charged chiral molecules into the system, the authors form transient chiral supramolecular structures, that decay automatically upon standing a few hours without shaking. Moreover, authors produced transient florescent patterns using ultrasound waves to control oxygenation. The work is well-presented and sound, however this reviewer has significant novelty concerns.

Major concerns:

1) The assembly between the same viologen derivative and pyranine has been already described, see *Chem. Commun.*, 2017,53, 2371-2374. This work is also not cited in the article, which raises ethical concerns.

2) Hwang et al. reported in *Nat. Chem.* 12, 808–813 (2020) the generation of reproducible spatiotemporal patterns of viologens using oxygen in the presence of sound waves, which is exactly the same strategy used here.

Minor:

- Page 3, main text: the EPR signal is indicated at 387 mT, but in the EPR spectrum showed in Figure S8 the highest value is at 345 mT, please clarify.

- Page 10, main text: the EPR signal is reported at 379 mT, but in the EPR spectrum showed in Figure S14 the highest value is at 345 mT, please clarify.

- Also at page 10 there seems to be a discrepancy regarding the concentration of said sample: in the main text is reported C12-MV2+/ PN (1 mM), in the supporting info instead, C12-MV2+/ PN (3 mM)

Reviewer #2:

Remarks to the Author:

The authors have merged two concepts - dissipative self-assembly and mechanical forces to fabricate systems where these two aspects are integrated, giving rise to non-equilibrium supramolecular polymerisations. The authors made use of reversible chemical reconfigurations in aqueous media based on redox responsiveness. Mechanical agitation (shaking) was used to direct the oxidation of viologens followed by self-assembly of pyranine, demonstrating disassembly over time in the presence of reducing agent. Interestingly, the generality of their approach was applied to self-assembling systems, engineering chiral charged supramolecular helical structures. Moreover, the use of ultrasound, as an alternative source of vibration was utilised to generate reproducible patterns, which could be emerged from oxygen rich and oxygen poor chemical domains. One of the most important aspects of the work is that nitrogen gas, which acts as the side product leaves the solution, giving rise to traceless waste in the overall process. The non-equilibrium pathway was monitored using a variety of spectroscopic and microscopic techniques, where exposure to mechanical agitation and ultrasound led to dynamic formation of aggregates with controllable lifetimes.

The design and construction of dissipative supramolecular systems has been an intensive area of research during the last decade, where waste accumulation is considered as one of the major challenges. Yet, overall, the type of non-equilibrium systems that minimize waste formation remain limited. The results are in line with expectations in terms of reaction dynamics and lifetime of assemblies formed, as described and cited e.g. by Jan Van Esch (*Angew. Chem. Int. Ed.*, 2010, 49, 4825-4828) and Hermans (*Nat. Nanotechnol.*, 2018, 13, 1021-1027). Given that there are only a couple of reports (*Nat. Chem.*, 2010, 2, 977-983, *Chem*, 2022, 8, 2063-2065) on the effect of low-

frequency sound waves on self-assembly, the integration of fuelling-driven complexification with shaking to achieve waste-free non-equilibrium responses is the most exciting part. The approach taken here could open up new perspectives in the fabrication of active and acoustic materials. In general, the results support the conclusions, however, more experimental evidence and additional control experiments are required, with some suggestions provided below.

Structure and references: The authors discussed in the introduction about non-equilibrium self-assembly, focusing primarily on the use of chemical and other type of fuels to trigger dynamic supramolecular reconfigurations. I believe that it would be beneficial for the readers to have in the introduction a discussion about sound-induced dissipative self-assembly and what could potentially be the key differences of this work compared to Kim's recently published papers on the use of audible sound. This only appears at the end of the manuscript, on the section of ultrasound induced pattern formation. In this way, the authors can introduce early on differences in terms of frequencies (audible sound, ultrasound and shaking), in which these systems might respond to and dynamically adapt. Moreover, the work from Feringa and co-workers on mechanically induced gel formation (Langmuir 2013, 29, 28, 8763-8767) should also be cited and discussed.

Comments/Questions

1) The authors discuss about critical aggregation effects of the molecules involved in the self-assembling systems only in the supporting information. It will be beneficial to introduce these effects in the manuscript and emphasize on the importance of concentration on the resulting self-assembling structures. Are there any differences in terms of morphology and kinetics if the shaking is introduced at higher concentration of the building blocks in the mixtures? These effects have been observed before in mechanically induced organogel formation (Langmuir 2013, 29, 28, 8763-8767). Moreover, the concentration for the viscosity experiments involving C12-MV2+ and PN was 3 mM, while for TEM measurements was 1 mM each. Is this difference arises from dilution effects to better visualise the structures for the microscopy measurements?

2) Regarding the transient supramolecular helical structures in the presence of L and D- Phenyllactic acid - the CD spectra showed for the L-derivative red-shifted structures upon letting the samples to relax (stay) and blue-shifted structures for the D-derivative. Are there different structures involved as the system gradually relaxes back to the original state? Given that these changes might emerge in solution, Cryo-EM could be helpful to support the dynamic transitions (if any).

3) In the ultrasound-induced pattern formation, do the authors exclude any contribution from cavitation effects? As the samples are exposed to ultrasound for 2.5 minutes, the formation of high energy pockets might affect the lifetime of the structures formed and the fabrication of the patterns. Are these effects contributing less compared to the effect of diffusion and dissolution of gases?

4) The patterns which are formed by ultrasound are quite interesting, where the effect of viscosity and shape of the vibrating dish has been shown before to impact the local domains. Are the heart-shape patterns reconfigurable? Can the authors obtain different patterns by increasing the sonication time, or by sequentially exposing the systems to ultrasound and shaking?

5) The authors should give the exact details on sample preparation for the different systems: i) which buffer was used, ii) what is the concentration of the buffer, iii) how maybe the pH dropped during the experiments, iv) for how long the systems were agitated?

Reviewer #3:

Remarks to the Author:

This manuscript reports a non-equilibrium supramolecular polymerization using oxygen and hydrazine as chemical fuels. Although similar types of supramolecular polymerizations have been reported a lot

recently, almost all of them were hampered by the generated chemical wastes. On the other hand, in the reported system, the wastes are converted into inert N₂ gas, and the authors demonstrated that the non-equilibrium supramolecular polymerization could be repeated many times as far as chemical fuels are present. This is significant progress in the field of supramolecular polymerization. Furthermore, the authors demonstrated unique behaviors of the system by combining ultrasound. These additional efforts also gave me a good impression. The authors supported their claims well with experiments, and I recommend the publication of this manuscript in Nature Communications after the following minor revisions.

(1) In Fig.2(b), the authors discussed the degree of aggregation based on viscosity. Here, the unit of the y-axis is %. How do the authors define 100% of aggregation? In this case, to define 100% of aggregation with viscosity, the authors must prove that this system is now in a 100%-aggregation state by other methods. Otherwise, the authors cannot know the viscosity of a 100%-aggregation-state.

(2) The authors should discuss the destination of O₂ in this system. If possible, the authors should prove it through experiments. According to Fig.1a, O₂ is converted into O₂⁻. This may react with N₂H₄ under basic conditions and generate NO or other chemicals (I am not sure at all). If the authors characterize the gas phase of the closed system by gas chromatography, the authors might be able to discuss the destination of O₂. The destination of O₂ is essential for the main claim of this manuscript. The authors should at least provide a reasonable assumption on it.

Reviewer #1 (Remarks to the Author):

The authors report the transient self-assembly of alkyl substituted viologen and pyranine using oxygen and hydrazine hydrate as oxidizing and reducing agent, respectively. By shaking the system, they accelerated the diffusion of oxygen from air into solution, resulting in oxidation of viologen and co-assembly. As shaking is stopped, viologen gradually reduced by hydrazine, leading to disassembly. Introducing charged chiral molecules into the system, the authors form transient chiral supramolecular structures, that decay automatically upon standing a few hours without shaking. Moreover, authors produced transient florescent patterns using ultrasound waves to control oxygenation. The work is well-presented and sound, however this reviewer has significant novelty concerns.

Reply: We express our sincere gratitude to the reviewer for the valuable comments and feedback. Upon careful consideration, we acknowledge that the innovative nature of our work was not adequately highlighted in the previous manuscript. Therefore, we have taken the reviewer's comments into account and made significant revisions to the original manuscript, particularly in the introduction (paragraph 3) and the discussion, to articulate our approach.

Major concerns:

The assembly between the same viologen derivative and pyranine has been already described, see *Chem. Commun.*, 2017,53, 2371-2374. This work is also not cited in the article, which raises ethical concerns.

Reply: Thanks for pointing out the missing reference, which has been cited in the revised manuscript. Moreover, beyond the viologen-pyranine based supramolecular system, we also proved our concept, mechanosensitive non-equilibrium supramolecular polymerization, can be extended to other types of redox reaction based systems. We designed and synthesized a naphthalenetetracarboxylic diimide (NDI) based supramolecular system. As we expected, the NDI-based system also showed clearly mechanosensitive non-equilibrium polymerization phenomena. We added one paragraph and one figure (Figure 6) to describe the new data in the main text. The synthesis and characterization of amphiphilic compound C₁₂-NDI-PEG350, which was prepared by a statistical condensation reaction of 1,4,5,8-naphthalenetetracarboxylic dianhydride with hydrophilic polyethylene glycol groups amine and hydrophobic dodecyl amine, were in the supporting information in the revised version (Scheme 5, Figure S54 and S55).

Scheme 5. Synthesis of C₁₂-NDI-PEG350.

Figure S54. ¹H-NMR spectrum of C₁₂-NDI-PEG350 in CDCl₃.

Figure S55. ¹³C-NMR spectrum of C₁₂-NDI-PEG350 in CDCl₃.

Figure 6. Shake-induced out-of-equilibrium redox reaction of C_{12} -NDI-PEG350 in a closed system. (a) Schematic representation of dissipative supramolecular polymerization based on C_{12} -NDI-PEG350 triggered by shake. (b) Visualization of chemochromism of the solution indicating the shake-induced transient redox reaction, $[C_{12}\text{-NDI-PEG350}] = 1 \text{ mM}$, $[\text{N}_2\text{H}_4\cdot\text{H}_2\text{O}] = 10 \text{ \% (v/v)}$. (c) Time-dependent UV-vis spectra depicting redox behavior of C_{12} -NDI-PEG350 (1 mM) triggered by shaking in a buffer solution (pH = 8) containing reducing agent ($\text{N}_2\text{H}_4\cdot\text{H}_2\text{O}$, 10% v/v). (d, e, f, g, h) TEM images of solution C_{12} -NDI-PEG350 (1 mM) with the presence of $\text{N}_2\text{H}_4\cdot\text{H}_2\text{O}$ (10% v/v) before and after shaking over time.

2) Hwang et al. reported in *Nat. Chem.* 12, 808–813 (2020) the generation of reproducible spatiotemporal patterns of viologens using oxygen in the presence of sound waves, which is exactly the same strategy used here.

Reply: We appreciate the questions raised by the reviewer, as they have aided in our contemplation of the innovative aspects of this project. Audio-speaker and ultrasonic equipment both generate sound wave, however, there are several different points when low-frequency acoustic waves and ultrasound are used separately as a trigger for the generation of reproducible spatiotemporal patterns of viologens. (1) Patterns different: regular round circles in Hwang's work (based on low-frequency sound wave vibrations) while regular heart shape in our work (based on ultrasonic equipment). (2) Proposed mechanism is different: patterning formation is due to different concentrations of oxygen at the antinodal and nodal positions on the water surface based on audio-speaker (This mechanism was proposed by Kim et al in *J. Chem. Educ.* 2022, 99, 1539–1544; the formation of pattern is a replication of the vibrational ripples on the surface of the solution, Figure S34a and S35a), while the patterning formation under ultrasonic mode is due to the induced directional diffusion of oxygen, the diffusion direction is shown in Support Information Figure S35c and S36. We supplemented new photos and videos to demonstrate this diffusion phenomenon. It is clear from Video S7 that the formation of the heart-shaped pattern is a result of its diffusion behavior. (3) Viscosity dependence is different: heart-shaped patterns under ultrasound can be resized, unlike regular circles in low-frequency sound waves which are not adjustably resizable by altering the viscosity (Figure S37). We have observed that increasing viscosity can enhance the continuity of ultrasonically induced patterns and enable better control over the size of heart-shaped patterns. According to Newton's law of viscosity, the magnitude of viscosity is proportional to the magnitude of shear stress (internal friction). With the power of ultrasound being constant and the driving force unchanged, an increase in resistance reduces the range of influence of central diffusion flow on the surrounding solution. We added related discussion in the revised main text.

Figure S34. (a) Regular vibrational pattern on the surface of the solution under low-frequency vibrations, as displayed by light reflection. (b) Spatiotemporal pattern

formed under low-frequency vibrations. (c) Chaotic vibrational pattern on the surface of the solution under ultrasonic vibrations, as displayed by light reflection. (d) Spatiotemporal pattern formed under ultrasonic vibrations.

Figure S35. Proposed mechanisms of sound-induced patterns. (a) Schematic of the experimental setup for audible sound-controlled pattern generation experiment (left). Schematic cross-sectional view of region-specific dissolution of gases into a vertically vibrating solution (right). (b) Schematic illustration of ultrasound-induced oxygen directed diffusion to form patterns.

Figure S36. The images show the diffusion direction of O_2 during patterning induced by ultrasound. (a) Ultrasound-induced patterning of MV^{2+} (5 mM), PEG (10 kDa, 5%). (b) Ultrasound-induced patterning of $C_{12}-MV^{2+}$ (5 mM). Obtained by sonication for 150 s and stay for 60 s. $N_2H_4 \cdot H_2O$ (20% v/v).

Figure S37. Photographs depicting the effect of varying viscosities on pattern formation in the presence of low frequency acoustic vibrations (60Hz, audio-sound). MV^{2+} (5 mM) with different content of PEG (10 kDa), $N_2H_4 \cdot H_2O$ (20% v/v).

Figure S28. Photographs depicting the effect of varying viscosities on pattern formation triggered by high frequency acoustic vibrations (40 kHz, ultrasound). MV^{2+} (5 mM) with different content of PEG (10 kDa), $N_2H_4 \cdot H_2O$ (20% v/v).

Minor:

Page 3, main text: the EPR signal is indicated at 387 mT, but in the EPR spectrum showed in Figure S8 the highest value is at 345 mT, please clarify.

Reply: Thanks for pointing out this mistake. We corrected the error and changed “387 mT” to “345 mT” in the revised manuscript.

- Page 10, main text: the EPR signal is reported at 379 mT, but in the EPR spectrum showed in Figure S14 the highest value is at 345 mT, please clarify.

Reply: Thanks for pointing out this mistake. We corrected the error and changed “379 mT” to “345 mT” in the revised manuscript.

- Also at page 10 there seems to be a discrepancy regarding the concentration of said sample: in the main text is reported $C_{12}-MV^{2+}/PN$ (1 mM), in the supporting info instead, $C_{12}-MV^{2+}/PN$ (3 mM).

Reply: Thanks for pointing out this mistake. We corrected the error and changed “3 mM” to “1 mM” in the revised supporting information.

Reviewer #2 (Remarks to the Author):

The authors have merged two concepts - dissipative self-assembly and mechanical forces to fabricate systems where these two aspects are integrated, giving rise to non-equilibrium supramolecular polymerisations. The authors made use of reversible chemical reconfigurations in aqueous media based on redox responsiveness. Mechanical agitation (shaking) was used to direct the oxidation of viologens followed by self-assembly of pyranine, demonstrating disassembly over time in the presence of reducing agent. Interestingly, the generality of their approach was applied to self-assembling systems, engineering chiral charged supramolecular helical structures. Moreover, the use of ultrasound, as an alternative source of vibration was utilised to generate reproducible patterns, which could be emerged from oxygen rich and oxygen poor chemical domains. One of the most important aspects of the work is that nitrogen gas, which acts as the side product leaves the solution, giving rise to traceless waste in the overall process. The non-equilibrium pathway was monitored using a variety of spectroscopic and microscopic techniques, where exposure to mechanical agitation and ultrasound led to dynamic formation of aggregates with controllable lifetimes.

The design and construction of dissipative supramolecular systems has been an intensive area of research during the last decade, where waste accumulation is considered as one of the major challenges. Yet, overall, the type of non-equilibrium systems that minimize waste formation remain limited. The results are in line with expectations in terms of reaction dynamics and lifetime of assemblies formed, as described and cited e.g. by Jan Van Esch (*Angew. Chem. Int. Ed.*, 2010, 49, 4825-4828) and Hermans (*Nat. Nanotechnol.*, 2018, 13, 1021-1027). Given that there are only a couple of reports (*Nat. Chem.*, 2010, 2, 977-983, *Chem*, 2022, 8, 2063-2065) on the effect of low-frequency sound waves on self-assembly, the integration of fuelling-driven complexification with shaking to achieve waste-free non-equilibrium responses is the most exciting part. The approach taken here could open up new perspectives in the fabrication of active and acoustic materials. In general, the results support the conclusions, however, more experimental evidence and additional control experiments are required, with some suggestions provided below.

Reply: We thank the Reviewer for the positive assessment of our work.

Structure and references: The authors discussed in the introduction about non-equilibrium self-assembly, focusing primarily on the use of chemical and other type of fuels to trigger dynamic supramolecular reconfigurations. I believe that it would be beneficial for the readers to have in the introduction a discussion about sound-induced dissipative self-assembly and what could potentially be the key differences of this work compared to Kim's recently published papers on the use of audible sound. This only appears at the end of the manuscript, on the section of ultrasound induced pattern formation. In this way, the authors can introduce early on differences in terms of frequencies (audible sound, ultrasound and shaking), in which these systems might

respond to and dynamically adapt. Moreover, the work from Feringa and co-workers on mechanically induced gel formation (Langmuir 2013, 29, 28, 8763-8767) should also be cited and discussed.

Reply: We are grateful for the valuable advice provided by the reviewer, whose insightful suggestions have greatly improved the quality of our work. We have carefully incorporated the relevant discussion and citations into the revised manuscript, which we believe has considerably strengthened the overall coherence and impact of our manuscript.

Comments/Questions

1) The authors discuss about critical aggregation effects of the molecules involved in the self-assembling systems only in the supporting information. It will be beneficial to introduce these effects in the manuscript and emphasize on the importance of concentration on the resulting self-assembling structures. Are there any differences in terms of morphology and kinetics if the shaking is introduced at higher concentration of the building blocks in the mixtures? These effects have been observed before in mechanically induced organogel formation (Langmuir 2013, 29, 28, 8763-8767). Moreover, the concentration for the viscosity experiments involving C₁₂-MV²⁺ and PN was 3 mM, while for TEM measurements was 1 mM each. Is this difference arises from dilution effects to better visualise the structures for the microscopy measurements?

Reply: We greatly appreciate the suggestions provided by the reviewer. We added additional discussion on the critical aggregation effects of molecules for the self-assembling systems in revised manuscript. Below is the relevant description: we constructed the mechanosensitive supramolecular self-assembly system based on the study of thermodynamically stable supramolecular polymerization of alkane substituted viologen (C₁₂-MV²⁺) and pyranine (PN), in which the self-assembly was driven by charge transfer interaction (CT interaction) and amphiphilic interaction. To investigate the self-assembly morphology of supramolecular polymers, firstly, we performed viscosity studies on different ratios and concentrations of C₁₂-MV²⁺/PN samples. The viscosity was dependent on the molar ratio between C₁₂-MV²⁺/PN (Figure 2) and concentration of monomers (Figure S6). TEM experiment also clearly proved that the assembly was in line with our previous assumption. When 0.5 eq PN was added to the C₁₂-MV²⁺ solution (1 mM), it exhibits a remarkable aggregation effect, whereas the critical micelle concentration (CMC) of pure C₁₂-MV²⁺ solution is 3.2 mM. The above results suggest that supramolecular polymer of C₁₂-MV²⁺/PN is a structure control polymerization.

Figure 2. Characteristics of structure-controlled supramolecular polymerization. (a) Schematic diagram of structure-controlled self-assembly dependent on the ratios of $C_{12}\text{-MV}^{2+}$ and PN. (b) Viscosity of different ratios of $C_{12}\text{-MV}^{2+}$ and PN solutions. TEM images depicting the self-assembly morphology resulting from varying amounts of PN added to $C_{12}\text{-MV}^{2+}$ solution, 0.5 mM PN (c), 1 mM (d), and 1.5 mM (e). $[C_{12}\text{-MV}^{2+}] = 1 \text{ mM}$.

Figure S4. TEM images of molar equivalent $C_{12}\text{-MV}^{2+}/\text{PN}$ at different concentrations. (a) 0.5 mM; (b) 1 mM; (c) 3 mM; (d) 5 mM.

Figure S5. Respective TEM images show the transformation from micelles to fiber collected at 0.5 equiv, 1 equiv, and 1.5 equiv of PN against $C_{12}\text{-MV}^{2+}$ (1 mM).

Figure S6. Viscosity of the solution at different concentrations.

Thanks for pointing out the missing reference. Feringa's work is very instructive and helpful, and it has been cited in the revised manuscript. Additionally, to investigate the effect of concentration on the self-assembled structure, we first performed TEM analysis on molar equivalent $C_{12}\text{-MV}^{2+}$ /PN at different concentrations (0.5 mM, 1 mM, 3 mM, and 5 mM). The images revealed that molar equivalent $C_{12}\text{-MV}^{2+}$ and PN self-assembled into nanofibers, with the number of fibers increasing as the concentration increased (Figure S4), which could be further corroborated by the results of viscosity tests (Figure S6). Therefore, for better observation, we selected a concentration of 1 mM and 3 mM for TEM analysis (Figure S15 and Figure 2). The morphologies and transient self-assembly kinetics of samples of $C_{12}\text{-MV}^{2+}$ and PN on (1 mM/1 mM) and (3 mM/3 mM) are similar, as seen from TEM (Figure 2, Figure S15). The difference point is the density of formed tubes is higher when the concentration is high.

Figure S15. (a-f) TEM images of solution with $C_{12}\text{-MV}^{2+}$ /PN (3 mM/ 3 mM) in the presence of $\text{N}_2\text{H}_4\cdot\text{H}_2\text{O}$ (10% v/v) before and after shake over time.

2) Regarding the transient supramolecular helical structures in the presence of L and D-Phenyllactic acid - the CD spectra showed for the L-derivative red-shifted structures upon letting the samples to relax (stay) and blue-shifted structures for the D-derivative. Are there different structures involved as the system gradually relaxes back to the original state? Given that these changes might emerge in solution, Cryo-EM could be helpful to support the dynamic transitions (if any).

Reply: Thanks to the reviewer for pointing out the question on “blue shift” or “red shift” issue. Firstly, we have investigated the time-dependent UV-vis spectra of shake-driven transient supramolecular helical structure in the presence of L- phenyllactic acid and D-phenyllactic acid, respectively. As shown in Figure S18, we did not observe a significant “red shift” or “blue shift” in the spectra. We re-recorded the CD spectra of chiral molecular induced transient supramolecular helical polymers. It turned out it was random to show a “red” or “blue” shift in the CD spectra, which might be the result of

the unstable self-assembled structures under the non-equilibrium state.

Figure S18. Time-dependent changes in UV-vis spectra demonstrating shake-driven a temporal helical polymer in the presence of (a) *L*-(-)-Phenylactic acid or (c) *D*-(+)-Phenylactic acid ($N_2H_4 \cdot H_2O$, 5% v/v). (b) The maximum absorbance wavelength change of shake driven transient polymer over time in the presence of (b) *L*-(-)-Phenylactic acid or (d) *D*-(+)-Phenylactic acid.

3) In the ultrasound-induced pattern formation, do the authors exclude any contribution from cavitation effects? As the samples are exposed to ultrasound for 2.5 minutes, the formation of high energy pockets might affect the lifetime of the structures formed and the fabrication of the patterns. Are these effects contributing less compared to the effect of diffusion and dissolution of gases?

Reply: We greatly appreciate the reviewer's question as it has expanded our thoughts on the pattern formation process under research. The cavitation effect of ultrasound promotes heterogeneous reactions by aiding in the uniform mixing of heterogeneous reactants and accelerating the diffusion of reactants and products, which to some extent, facilitates the oxidation process. Due to the ease with which the reduced free radicals are oxidized by oxygen, resulting in a color change, we think that the pattern formation resulting from ultrasound is primarily due to its diffusion behavior. (For detailed

information about the proposed mechanism of pattern formation, please refer to the response to the Reviewer#1.) Although locally high energy pockets to some extent facilitate the oxidation process, the oxidation rate is not the main focus of pattern formation. We have added a new video S7 to demonstrate the directional diffusion phenomenon during the ultrasound process.

4) The patterns which are formed by ultrasound are quite interesting, where the effect of viscosity and shape of the vibrating dish has been shown before to impact the local domains. Are the heart-shape patterns reconfigurable? Can the authors obtain different patterns by increasing the sonication time, or by sequentially exposing the systems to ultrasound and shaking?

A: Thanks for the reviewer's question. Yes, the heart-shaped pattern generated by ultrasound can be reproducible. In the ultrasonic patterning cycle, as long as the fully oxidized solution is sealed and left to stand, it can return to its initial state and continue to the next cycle until the reducing agent is depleted as shown in Figure S32. At the same time, the heart pattern can be reproduced under different solution compositions and different lighting conditions as shown in Figure S28. We have supplemented the experiment by using a new switchable frequency ultrasound device to demonstrate this point. The methyl violet free radical solution can produce similar heart-shaped patterns at 28kHz and 40kHz, and the pattern can be reproduced on different ultrasound devices (Figure S39). By increasing the ultrasound time, patterns in different diffusion stages can be obtained, as shown in Figure S38 and video S7. Since irregular shaking can easily destroy the pattern formed by ultrasound, we attempted to combine regular low-frequency vibration with ultrasound. However, due to the different imaging principles of these two methods, ultrasound can easily destroy the regular pattern formed by low-frequency vibration, forming some irregular patterns, as shown in Figure S40. (For detailed information about the causes of patterning, please refer to the response to the second inquiry of the Reviewer#1.) Nevertheless, we have tried ultrasound vibrations of different frequencies, which have formed similar heart-shaped patterns, only with slight differences in size (Figure S39).

Figure S32. Reproducible ultrasound-induced patterns for MV^{2+}/PN in buffer containing 5% PEG, under 365 nm UV light.

Figure S28. Ultrasound-induced patterns, MV^{2+} under daylight (Left), MV^{2+}/PN under daylight (Middle), MV^{2+}/PN under 365 nm UV light (Right). Solution containing 5% PEG (10 kDa).

Figure S38. Photographs showing fluorescence patterns generation over time induced by continuous ultrasound (40 kHz), under 365 nm UV light. MV^{2+}/PN (5 mM/ 5 mM) in buffer containing 5% PEG (10kDa).

Figure S39. Changes in patterns obtained at different ultrasound frequencies. MV^{2+} (5 mM) solution with 2.5% PEG (10 kDa).

Figure S40. Ultrasound destroys the regular pattern formed by low frequency vibrations (40 Hz). MV^{2+} (5 mM) solution with 2.5% PEG (10 kDa).

5) The authors should give the exact details on sample preparation for the different systems: i) which buffer was used, ii) what is the concentration of the buffer, iii) how maybe the pH dropped during the experiments, iv) for how long the systems were agitated?

Reply: We appreciate these suggestions from the reviewer and have made revisions to the supporting material to include additional details on the preparation of the buffer solutions and sample preparation. Please see below for specific descriptions:

(1) Preparation of buffer solutions:

Boric acid-potassium chloride-sodium hydroxide buffer solution (pH 8) was prepared by mixing 25 mL of boric acid-potassium chloride (0.2 M) with 4 mL of 0.1 M aqueous sodium hydroxide and then diluting the mixture to 100 mL with water.

Ammonium chloride-ammonia buffer solution (pH 9.18) was prepared by mixing 0.1 mol/L ammonium chloride with 0.1 mol/L ammonia in a 2:1 ratio.

Boric acid-potassium chloride-sodium hydroxide buffer solution (pH 10) was prepared by mixing 25 mL of boric acid-potassium chloride with 43.9 mL of 0.1 M aqueous sodium hydroxide and then diluting the mixture to 100 mL with water.

Disodium hydrogen phosphate-sodium hydroxide buffer solution (pH 12) was prepared by mixing 50 mL of 0.05 M disodium hydrogen phosphate solution with 26.9 mL of 0.1 M aqueous sodium hydroxide solution and then diluting the mixture to 100 mL with water.

Potassium chloride-sodium hydroxide buffer solution (pH 13) was prepared by mixing 25 mL of 0.2 M potassium chloride solution with 66 mL of 0.2 M aqueous sodium hydroxide solution and then diluting the mixture to 100 mL with water.

To monitor the change in solution pH during the experiment, we used a pH meter to measure the pH of buffer solution (pH 12) containing $C_{12}-MV^{2+}$ (3 mM) and PN (3 mM) in the presence of $N_2H_4 \cdot H_2O$ (5% v/v) over time. The results indicated that the pH of the buffer solution remained constant throughout the experiment.

(2) Sample preparation

The tests of a shake-driven non-equilibrium system of $C_{12}-MV^{2+}$ were carried out in a buffer solution with pH 12, and the tests for $C_{12}-DNI-PEG350$ were performed in a pH 8 buffer solution. As an example, a shake-driven transient supramolecular polymerization consisting of $C_{12}-MV^{2+}$ and PN was used. Stock solutions of $C_{12}-MV^{2+}$

(5 mM) and PN (10 mM) were separately prepared in the buffer solution with a pH of 12. To prepare the sample, the required amount of stock solution of $C_{12}\text{-MV}^{2+}$ was added to a bottle and diluted to the desired concentration with phosphate buffer (pH 12). Afterward, the required quantities of PN stock solution and hydrazine hydrate were added successively to the above solution under vortex conditions, followed by sonication for 1 minute. The solution was then left to stand for a while until complete reduction of $C_{12}\text{-MV}^{2+}$ to $C_{12}\text{-MV}^{+}$ occurred before subsequent testing.

(3) Ultrasound-induced pattern generation experiments

The experiment utilized the MV^{2+} -PN as a model system to demonstrate ultrasound-induced transient patterning. Initially, MV^{2+} , PN, and PEG were dissolved in a pH 12 buffer solution within a 50 ml centrifuge tube. Subsequently, the solution was supplemented with hydrazine hydrate and thoroughly mixed. After MV^{2+} was fully reduced to MV^{+} , the solution was poured into a mold in the sonication apparatus and then a distinct fluorescent heart-shaped pattern appeared after 150 s of sonication and about 60 s of resting, under 365 nm ultraviolet light. Afterward, sonication was continued to allow complete oxidation of the sample. Finally, leaving the solution sealed for 30 min would return it to its initial state, while leaving it in the open state would result in a random spot pattern.

We hope that these revisions provide clarity and help to ensure an accurate recording of our experimental procedures. Thank you again for your valuable feedback.

Reviewer #3 (Remarks to the Author):

This manuscript reports a non-equilibrium supramolecular polymerization using oxygen and hydrazine as chemical fuels. Although similar types of supramolecular polymerizations have been reported a lot recently, almost all of them were hampered by the generated chemical wastes. On the other hand, in the reported system, the wastes are converted into inert N₂ gas, and the authors demonstrated that the non-equilibrium supramolecular polymerization could be repeated many times as far as chemical fuels are present. This is significant progress in the field of supramolecular polymerization. Furthermore, the authors demonstrated unique behaviors of the system by combing ultrasound. These additional efforts also gave me a good impression. The authors supported their claims well with experiments, and I recommend the publication of this manuscript in Nature Communications after the following minor revisions.

Reply: Thanks to the reviewer for the positive comment!

(1) In Fig.2(b), the authors discussed the degree of aggregation based on viscosity. Here, the unit of the y-axis is %. How do the authors define 100% of aggregation? In this case, to define 100% of aggregation with viscosity, the authors must prove that this system is now in a 100%-aggregation state by other methods. Otherwise, the authors cannot know the viscosity of a 100%-aggregation-state.

Reply: Thanks for pointing out this question and the assertion of 100%-aggregation-state is not rigorous enough. Drawing on George's study (Nat Commun, 2019, 10, 450), the extent of self-assembly of C₁₂-MV²⁺/PN is defined as α , which could be confirmed according to Equation (1). As shown in Figure 3b and 3c, the previous statement has been replaced with numerical values of '1' and '0', representing the highest and lowest degree of aggregation, respectively.

$$\alpha = \frac{\eta - \eta_0}{\eta_{max} - \eta_0} \quad (1)$$

α ---the degree of aggregation; η_{max} --- viscosity at maximum aggregation; η ---viscosity at different aggregation degrees; η_0 ---initial viscosity.

Figure 3. (b) Time-dependent evolution of viscosity trend depicting the kinetics of transient supramolecular polymerization, C₁₂-MV²⁺ (3 mM) and PN (3 mM) in the presence of N₂H₄•H₂O (10% v/v). (c) Polymerization-depolymerization cycles of MV²⁺ and PN driven by shake. α is the extent of self-assembly of C₁₂-MV²⁺/PN.

Additionally, a viscosity analysis confirmed that the molar equal C₁₂-MV²⁺/PN (3

mM) specimens have achieved the highest viscosity corresponding to the maximum degree of aggregation (Figure 2b). What's more, TEM images further confirmed molar equivalent $C_{12}\text{-MV}^{2+}$ and PN (3 mM) self-assembled into well-defined nanofibers with 80 nm in diameter and several hundreds of microns in length as shown in Figure 2c. Moreover, dynamic light scattering (DLs) was employed to determine the size of the assembly. As shown in Figure S8, the results indicated that the assembly reaches its maximum diameter (108 nm) only when the ratio of incoming PN to $C_{12}\text{-MV}^{2+}$ is 1.

In summary, our findings support a maximum degree of aggregation of the assembly when the ratio of incoming PN to $C_{12}\text{-MV}^{2+}$ is equal, which was confirmed by both viscosity studies and DLs measurements. Moreover, our TEM images revealed the potential effects of PN on the assembly's structure, highlighting the importance of optimizing the ratio of incoming PN to $C_{12}\text{-MV}^{2+}$ in future studies. Therefore, we have replaced the previous statement of 100% aggregated state with both '1' and '0' states, and have added the definition of the extent of self-assembly to the methods (paragraph 4).

Figure 2. Characteristics of structure-controlled supramolecular polymerization. (a) Schematic diagram of structure-controlled self-assembly dependent on the ratios of $C_{12}\text{-MV}^{2+}$ and PN. (b) Viscosity of different ratios of $C_{12}\text{-MV}^{2+}$ and PN solutions. TEM images depicting the self-assembly morphology resulting from varying amounts of PN added to $C_{12}\text{-MV}^{2+}$ solution, 0.5 mM PN (c), 1 mM (d), and 1.5 mM (e). [$C_{12}\text{-MV}^{2+}$] = 1 mM.

Figure S8. Z-average diameter of C_{12} - MV^{2+} -PN aggregates in buffer solution (pH = 12), (a) $[PN] = 1$ mM, (b) $[C_{12}\text{-}MV^{2+}] = 1$ mM. DLs measurements of C_{12} - MV^{2+} -PN aggregates in buffer solution (pH = 12), (c) $[PN] = 1$ mM, (d) $[C_{12}\text{-}MV^{2+}] = 1$ mM.

(2) The authors should discuss the destination of O_2 in this system. If possible, the authors should prove it through experiments. According to Fig.1a, O_2 is converted into O_2^- . This may react with N_2H_4 under basic conditions and generate NO or other chemicals (I am not sure at all). If the authors characterize the gas phase of the closed system by gas chromatography, the authors might be able to discuss the destination of O_2 . The destination of O_2 is essential for the main claim of this manuscript. The authors should at least provide a reasonable assumption on it.

A: We express our utmost gratitude to the reviewer for suggesting that we examined the gas phase, which helped us understand the reaction process of the gas-liquid reaction. We used gas chromatography to monitor the gas composition during the redox process. Due to the low separation efficiency of gas chromatography columns, it was difficult to separate nitrogen and oxygen completely on GC, but we observed the general trend: a gradual decrease in oxygen and a progressive increase in nitrogen content over time (Figure S13b). To better track the kinetic changes in gas composition, as shown in Figure S13a, we employed an oxygen-nitrogen percentage detector to quantitatively

monitor the gas phase change process. The results obtained from the detector were consistent with those obtained from gas chromatography measurements. Additionally, we utilized detectors with ppm-level sensitivity for nitric oxide and nitrogen dioxide to confirm the absence of nitric oxide and nitrogen dioxide production.

Figure S13. (a) The composition of the gas detection equipment. (b) The gas chromatography spectrum displays the change process of oxygen and nitrogen. (c) The gas phase component changes in the redox process were quantitatively obtained through the gas detectors. [$C_{12-MV^{2+}}$] = 5 mM, $N_2H_4 \cdot H_2O$ (10% v/v), $V_g:V_l = 3:1$.

Reviewers' Comments:

Reviewer #2:

Remarks to the Author:

The authors have addressed all of my questions/comments with new experiments, and the new version of the manuscript has been substantially improved. This is also a result from addressing the comments from the other reviewers. Importantly and in line with comments from other reviewers, the novelty of the manuscript is better stretched in the new version, especially when it is compared with existing systems, making overall the message much clearer. I recommend publication in Nature Communications.

Reviewer #3:

Remarks to the Author:

The authors sufficiently answered to my questions/comments and revised manuscript properly. Hence, I would like to recommend the publication of this manuscript in Nature Communications.

Reviewer #2 (Remarks to the Author):

The authors have addressed all of my questions/comments with new experiments, and the new version of the manuscript has been substantially improved. This is also a result from addressing the comments from the other reviewers. Importantly and in line with comments from other reviewers, the novelty of the manuscript is better stretched in the new version, especially when it is compared with existing systems, making overall the message much clearer. I recommend publication in Nature Communications.

Reply: Thank you for your suggestion and evaluation. We are glad that you think our new version has been substantially improved and the novelty is better stretched. We look forward to publishing in Nature Communications.

Reviewer #3 (Remarks to the Author):

The authors sufficiently answered to my questions/comments and revised manuscript properly. Hence, I would like to recommend the publication of this manuscript in Nature Communications.

Reply: Thank you for your suggestion and evaluation. We are glad that you think we have sufficiently answered your questions/comments and revised the manuscript properly. We look forward to publishing in Nature Communications.